# Morphology, Micromechanical, and Macromechanical Properties of Novel Waterborne Poly(urethane-urea)/Silica Nanocomposites

**DOI:** 10.3390/ma16051767

**Published:** 2023-02-21

**Authors:** Veronika Gajdošová, Milena Špírková, Yareni Aguilar Costumbre, Sabina Krejčíková, Beata Strachota, Miroslav Šlouf, Adam Strachota

**Affiliations:** Institute of Macromolecular Chemistry, Czech Academy of Sciences, Heyrovskeho nam. 2, CZ-162 00 Praha, Czech Republic

**Keywords:** mechanical properties, microindentation, polyurethanes, nanocomposites, silica, aqueous polyurethane dispersion

## Abstract

Morphology, macro-, and micromechanical properties of novel poly(urethane-urea)/silica nanocomposites were analyzed by electron microscopy, dynamic mechanical thermal analysis, and microindentation. The studied nanocomposites were based on a poly(urethane-urea) (PUU) matrix filled by nanosilica, and were prepared from waterborne dispersions of PUU (latex) and SiO_2_. The loading of nano-SiO_2_ was varied between 0 (neat matrix) and 40 wt% in the dry nanocomposite. The prepared materials were all formally in the rubbery state at room temperature, but they displayed complex elastoviscoplastic behavior, spanning from stiffer elastomeric type to semi-glassy. Because of the employed rigid and highly uniform spherical nanofiller, the materials are of great interest for model microindentation studies. Additionally, because of the polycarbonate-type elastic chains of the PUU matrix, hydrogen bonding in the studied nanocomposites was expected to be rich and diverse, ranging from very strong to weak. In micro- and macromechanical tests, all the elasticity-related properties correlated very strongly. The relations among the properties that related to energy dissipation were complex, and were highly affected by the existence of hydrogen bonding of broadly varied strength, by the distribution patterns of the fine nanofiller, as well as by the eventual locally endured larger deformations during the tests, and the tendency of the materials to cold flow.

## 1. Introduction

The presented paper is dedicated to the study of the elastic and energy dissipation properties by micromechanical methods of nanocomposites based on an elastomeric polymer and a rigid filler, as well as to the comparison of the observed micro- and macromechanical properties (the latter determined by dynamic mechanical thermal analysis (DMTA)). For this purpose, the authors chose a recently prepared poly(urethane-urea)/silica nanocomposite with a highly regular and uniform filler [1], and they also took advantage of their experience with micro- and nanomechanical characterization of neat polymeric matrixes in different viscoelastic states [2,3].

**Polyurethanes (PUs),** which are used as matrices in the studied systems, belong to the most important mass-produced polymers [4,5]. A number of industrial and biochemical applications derive from their versatility. Initially developed to circumvent the patents covering polyamides (with which they share the feature of strong hydrogen bonding), polyurethanes have found a very wide application field, due to their ease of synthesis, and specific chemical (foaming by addition of water) and physical properties (mechanical strength, physical crosslinking, adhesion to biomaterials due to hydrogen bonds). Eventually, their application in foams (both flexible and rigid) became most important, but PUs are also produced as molded and cast massive materials, elastic textile fibers, lacquers, and adhesives. The mentioned polyurethane products are based on rigid thermoset networks, glassy or plastic thermoplastics, as well as PU elastomers [3,5].

Nanocomposites based on PU have been the subject of considerable research interest, which is largely due to the possibility of strong hydrogen bonding polymer–matrix. Examples include promising PU nanocomposite systems filled with clay [6], polyhedral oligomeric silsesquioxanes (POSS) [7], cellulose [8], TiO_2_ [9], Fe_3_O_4_ [10], or with nanosilica [11]. Nanofillers in general most frequently serve the purpose of providing very efficient mechanical reinforcement, due to their very high specific surface area [12]. In case of sufficiently small filler dimensions, optical transparency of the composite is preserved [13,14]. But nanofillers also are useful in lending specific optical [15,16], chemical [17,18,19,20,21], electrical [22], magnetic [23,24], or gas barrier [25,26] properties to the matrix.

Waterborne PU dispersions (wbPUD) have for a long time been the subject of considerable interest as replacements for organic solutions of PU, in the role of precursors of PU coatings [27], namely because of health, safety (flammability), and air pollution aspects. Different strategies of their preparation are reviewed and discussed in the literature [27,28].

The wbPUD are also an interesting starting material for the preparation of nanocomposites with hydrophilic fillers, such as nanosilica. An early route consisted in end-capping the PU polymer with coupling end groups (alkoxysilane) and in subsequent blending of the modified wbPUD with an aqueous suspension of nanosilica [29]. Other routes included the in-situ formation of nanosilica in an aqueous solution that was mixed with wbPUD [30], or the simple blending of commercial wbPUD with aqueous nanosilica [30,31]. The authors of the present work also employed the simple latter simple method for obtaining PU–silica nanocomposites in their earlier studies [1,32]. A sophisticated strategy for obtaining highly reinforced and resistant nanocomposites was based on the covalent fixation of the filler (commercial nanosilica) by pendant alkoxysilane groups of the wbPUD precursor [33]. The copolymeric PU matrix additionally was UV-cured in the final step. A similar route (with three variants) [34] was based on the use of wbPUD containing polymerizable vinyl groups, which were blended with commercial nanosilica at different stages of the multi-step synthesis of the wbPUD. The final step always consisted of the polymerization of the vinyl end groups while still in dispersion. Conceptually similar to the above, but not “waterborne”, was a nanocomposite synthesis based on blending in organic solution (dimethylacetamide), in which both the PU matrix and nanosilica were dispersed [35].

PUs based on polycarbonate as the polymeric diol component (PCD), which are the precursor of the “soft segments” that are responsible for elasticity [36,37,38,39], are attractive due to their resistance against weathering, fungi, and hydrolysis [37,39], as well as because of their biocompatibility [37,39]. The preparation of PCD-based PU coatings from a wbPUD precursor has been reported in [36,39]. In their recent work [40,41], the authors also studied this type of materials. The so obtained wbPCD-PUD displayed very good stability and storability (as dispersion) [41]: at least 9 months. The wbPCD-PUD system from [40] was further employed by the authors for the synthesis of PU–silica nanocomposites in [32]. Both the neat matrix and the nanocomposite were found to be fully recyclable via dissolution in acetone [32,40], but they possessed only modest mechanical properties. Hence, the authors improved their PCD-PU matrix (prepared via wbPUD) by using an excess of the diisocyanate component, which led to additional chain extension and to additional hydrogen bonding [41]. The improved matrix was used for synthesizing PU–silica nanocomposites, which then displayed excellent tensile properties [1]. The uniform shape and size of the nanofiller, as well as its high loadings, made the materials prepared in the very recent work [1] interesting for microindentation studies, which are in the focus of interest of the present work.

The mechanical and thermomechanical properties of polymers and their nanocomposites are standardly investigated using DMTA and/or tensile tests. In praxis however, situations often occur where only a few and small material specimens are available, which are not well-suited for fully-fledged (also statistically) DMTA or tensile characterization.

Microindentation techniques offer the advantage of easily probing tiny or irregular specimens, even fragments of mechanical parts or implants. Tens of indentations per specimen (even a small one) can be easily performed, so that a very high accuracy is achieved even with one small sample, in contrast to macroscopic mechanical testing. The microindentation techniques already are well-established in the field of hard inorganic materials, such as metal alloys and ceramics. In the case of polymers and their (nano)composites, indentation methods also are used, but less frequently, due to several problems: The first problem with such materials is the more complicated interpretation of the indentation results: The standard evaluation is mainly based on the Oliver & Pharr theory (O & P theory, see [42]), which works well for elastoplastic materials, but it is not certain how well it works for elastoviscoplastic ones, such as polymers and their composites, in which also the internal molecular friction can be prominent. The second problem with these types of samples is the fact that, depending on their characteristic glass transition temperature, their elastoviscoplastic character at room temperature can vary in a wide range, from soft and nearly ideally elastic materials (moduli in MPa range), via soft elastoviscous, through medium-hard elastoviscoplastic, to “very hard” glassy materials with predominant elastoplastic deformation (moduli in the GPa range). Hence, it is not entirely clear whether simple indentation methods will provide sufficient information for such a wide range of properties. The third, not yet fully resolved problem, is the question of at what filler loading (at a given constant temperature) does the character of the micro- and nanomechanical properties begin to change (between elastic, elastoviscous, etc.) [43,44]. Resolving the above-mentioned problems is a necessary condition for establishing the microindentation techniques as a standard method for characterizing the mechanical and thermomechanical properties of polymers and their (nano)composites, as an alternative to DMTA and tensile tests with large samples.

In a previous work [2], the authors investigated the ability of micro- and nanoindentation techniques to provide the full characterization (which conventionally is obtained by DMTA) of elasticity and energy dissipation-related properties, while studying specimens of simple crosslinked polymeric materials. It was possible to demonstrate that even quasistatic microindentation methods can supply information regarding both elastic stiffness and energy dissipation in such polymers. This was possible in fairly different material states, ranging from ideally glassy, via leather-like, up to ideally rubbery. In the simple samples studied in [2], the elasticity and energy dissipation was generated purely by the molecular properties of their polymer chains (theory: see [45]), namely by the enthalpy spring mechanism (caused by van der Waals forces) in the glassy state, by the entropy spring mechanism (coiling/uncoiling of polymer chains) in the rubbery state, by energy dissipation due to forcible conformational changes in the glassy or transitioning state, and by energy dissipation due to molecular friction in the rubbery state. The glass transition itself is caused by enabled/frozen conformational mobility above/below the glass transition temperature (see [45]).

In the present work, polymer nanocomposites are the focus of interest, with the aim to close the gap in knowledge concerning the possibility of their fully-fledged characterization by microindentation. In this context, the effect of filler–matrix interactions on elastic and energy dissipation-related properties, as determined by indentation, is of key importance.

To fulfill the above goals,

(1)We prepared a series of polymer nanocomposites with a (strongly hydrogen bonding) polycarbonate-based polyurethane matrix, reinforced by 0 to 40 wt% of monodisperse silica nano spheres; the nanocomposites were all formally in the rubbery state at room temperature, but their consistence varied from stiffer rubbery to semi-ceramic because of the filler effect.(2)We characterized the nanocomposites’ macromechanical properties using DMTA, and the micromechanical properties by microindentation hardness testing (MHI).(3)We compared the trends in elasticity and energy dissipation-related macro- and micromechanical properties.

## 2. Materials and Methods

### 2.1. Materials

The macrodiol T5652 (a polycarbonate diol (PCD)), with the number average molar mass (M_n_) of 2870 g × mol^−1^, is a short polymer end-capped by hydroxyl groups, which consists of the aliphatic spacers pentamethylene and hexamethylene (molar ratio of C5/C6 = 1:1), connected by carbonate groups. It was kindly donated by Asahi Kasei Chemical Corporation (Tokyo, Japan). 2,2-Bis(hydroxymethyl) propionic acid (DMPA; used as a chain extender and simultaneously as an ionization-able unit), 1,6-diisocyanatohexane (HDI; isocyanate component), and triethylamine (TEA; base) were purchased from Sigma-Aldrich (Burlington, MA, USA). Dried acetone (max. 0.0075% of H_2_O) was supplied by Merck KGaA (Darmstadt, Germany). The catalyst, dibutyltin dilaurate (DBTDL) from Sigma-Aldrich (Burlington, MA, USA) was supplied as a 10 wt% solution in Marcol oil (mixture of liquid saturated hydrocarbons). The nanofiller, Ludox AS 40, was also purchased from Sigma-Aldrich (Burlington, MA, USA): it is an aqueous dispersion (40 wt%) of silica nanospheres sized 22 to 34 nm, stabilized by ammonia adsorption, and displaying a zeta potential of −41 mV.

### 2.2. Preparation of PUU/Silica Composites

#### 2.2.1. Synthesis of Polyurethane-Urea Water Dispersions (PUUD)

The waterborne PUUD, which was used as the precursor of the matrix of the studied nanocomposites, was synthetized according to the same procedure that the authors developed in their previous work (see also Appendix A), where it is described in detail [1,41].

Molar ratios:

According to the mentioned procedure, DMPA acted simultaneously as a chain extender (because it is a short diol) and as ionizing agent (because it carries a carboxyl group). The molar ratio macro-diol (PCD type T5652) to DMPA was set equal to 1:1. The isocyanate to hydroxyl ratio (NCO:OH_total_) was set equal to 1.4:1 (where OH_total_ = OH_PCD_ + OH_DMPA_), which means a 40 mol% excess of isocyanate relatively to hydroxyl groups.

The synthesis of the dispersion consisted of three steps:(1)Pre-polymerization: PCD type T5652, DMPA, HDI, and DBTDL catalyst (0.05 mol% DBTDL per mol of NCO) were mixed in acetone at 60 °C and 700 rpm for 5 h.(2)Ionization via neutralization of the carboxylic groups in DMPA by adding triethylamine (TEA) to the acetone solution and mixing at 55 °C for 30 min.(3)Emulsification via gradual addition of water (water also reacts with the remaining NCO groups and thus connects the original polymer chains by newly formed urea, and eventually biuret groups). Finally, acetone was removed under reduced pressure and a purely waterborne PUUD, containing 30 ± 1 wt% of organic latex particles was obtained.

Synthesis details (component amounts) are given in the mentioned previous work (Table 1 of [41]), as composition “PUD 1.4/1:1”.

#### 2.2.2. Nanocomposite (PU/SiO_2_) Preparation via Solution Blending and Drying

The PU/silica nanocomposites were prepared from the waterborne PU latex dispersion (PUUD) and from an aqueous, ammonia-stabilized suspension of silica nanospheres (Ludox AS40, containing 40 wt% of silica in water). The procedure was the same as in the author’s previous work [1], and consisted of mixing and stirring the appropriate amounts of both dispersions (in order to obtain 5 to 40 wt% of silica in the dry nanocomposite) at room temperature for 1.5 h at 700 rpm, followed by two cycles of sonication in ultrasonic bath, each for 15 min.

For obtaining the final solid product (author’s procedure from [40]), the blended dispersion was finally cast into an open Teflon mold and left to slowly evaporate in air (5 days were needed). The so-obtained pre-dried films were subsequently warmed to 50 °C for 20 h in air, and finally vacuum-dried at 50 °C for 1 h (15 mbar). Continuous transparent or very slightly opaque films were obtained, which displayed a thickness of ca. 300 µm. As reference, a neat PU film (neat matrix) also was prepared, obtained by evaporating the neat PUUD dispersion.

Details of the procedure are given in the mentioned previous work [1,40].

The naming codes of the prepared nanocomposites (e.g., PUU05) refer to the wt% of silica (5 wt% in the PUU05 example) and are listed in Table 1.

### 2.3. Characterization of the Prepared PUU/Silica Nanocomposites

#### 2.3.1. Electron Microscopy

Dispersion of the silica nanofiller in the PU matrix was studied by a transmission electron microscopy (TEM; microscope Tecnai G2 Spirit Twin, FEI, Brno, Czech Republic) and scanning electron microscopy (SEM; microscope MAIA3, TESCAN, Brno, Czech Republic).

The specimens for TEM were prepared by ultramicrotomy. Ultrathin sections with a thickness of 60 nm were cut using an ultramicrotome with a cryo-attachment (Ultracut UCT, Leica, Vienna, Austria) using a diamond knife. The temperatures on the knife and samples were –40 °C and −70 °C, respectively. The ultrathin sections were visualized in the TEM microscope using bright-field imaging (TEM/BF) at the accelerating voltage of 120 kV.

The specimens for SEM were prepared by fracturing: small pieces of the samples were submerged in liquid nitrogen for 20 min and then broken using tweezers. Samples were fixed on a metallic support using conductive silver paste (Leitsilber G302, Christine Groepl, Austria) and covered with carbon in order to avoid charging and to increase the stability of the sample under the electron beam (C layer thickness 4 nm; vacuum evaporation device: JEE-4C, JEOL, Akishima, Tokyo, Japan). The fracture surfaces were visualized and analyzed in an SEM microscope MAIA3 (TESCAN, Brno, Czech Republic) equipped with an EDX detector (X-MaxN 20; Oxford Instruments, Abingdon, UK; SEM/EDX). The morphology was visualized at the accelerating voltage of 30 kV using both secondary electron detector (SEM/SE) and backscattered electron detector (SEM/BSE). The elemental composition was verified at the same accelerating voltage by means of energy-dispersive analysis of X-rays (SEM/EDX).

#### 2.3.2. Dynamic Mechanical Thermal Analysis (DMTA)

Dynamic mechanical properties of the nanocomposite products were tested with rectangular platelet samples, using an ARES G2 apparatus from TA Instruments (New Castle, DE, USA—part of Waters, Milford, MA, USA). The analyzed temperature range was from −65 to 100 °C, the heating rate 3 °C min^−1^. An oscillatory shear deformation at the constant frequency of 1 Hz was applied, whose amplitude was varied between 0.01 and 5% (regulated automatically by the auto-strain function, in response to sample resistance). The temperature dependences of the storage shear modulus (G’), of the loss modulus (G”) and of the loss factor tan(δ) were recorded. The standard specimen size was: 25 mm height, 10 mm width, and 0.3 mm thickness.

#### 2.3.3. Microindentation Hardness Testing

Micromechanical properties were obtained from instrumented microindentation hardness testing (MHI) performed with a microindentation combi tester (MCT tester; CSM Instruments SA, Peuseux, Switzerland). The MHI experiments were carried out using the Vickers method: a diamond square pyramid (with an angle of 136° between non-adjacent faces) was forced against the smooth flat specimen surface. The micromechanical properties are deduced from the dependence of the loading force on the indenter penetration depth.

The original surfaces of the PUU/silica nanocomposite films after preparation were found to be sufficiently flat and smooth for MHI measurements. Small pieces of samples (~1 × 0.5 cm) were fixed on a metallic support with a superglue. From each sample, at least 30 independent indentations were made and the final results were averaged. For each sample, we performed two sets of measurements with two different loading forces (100 mN and 200 mN), while the other experimental parameters (dwell time = time of maximal load = 6 s; and linear loading/unloading rate = 3000 mN/min) were unchanged.

Five micromechanical properties were obtained and evaluated from *F*-*h* curves (where *F* is the loading force and *h* is the indenter penetration depth) according to ISO 14577-1 standard: indentation modulus (*E*_IT_) is proportional to macroscopic elastic modulus; indentation hardness (*H*_IT_) is proportional to macroscopic yield stress; Martens hardness (*H*_M_), also known as the universal hardness; and indentation creep (*C*_IT_) is related to the macroscopic creep; and the elastic part of the indentation work (*η*_IT_) is defined as the ratio of the elastic deformation to total deformation. The exact definitions and detail information about the above-listed micromechanical properties can be found in our previous work [2,3,46] or in suitable reviews or textbooks dealing with micro- and/or nanoindentation [42,47]. The principle of MHI measurements is revised briefly in the Results section below (Section 3.2).

## 3. Results

The studied poly(urethane-urea)/silica nanocomposites were prepared in the same way as in our previous work [1] (see also Appendix A in the Appendix A), namely via the preparation of a stable aqueous PUU dispersion (PUUD), which enabled the subsequent easy blending of PUUD with a commercial aqueous suspension of silica nanospheres (Ludox). In the final preparation step, the aqueous mixture was dried to obtain the solvent-free nanocomposites. Standard specimens were obtained as films with the thickness of ~300 μm. The remarkable stability (and storability) of the employed PUUD dispersion (as well as of some related compositions) was characterized in detail (i.e. by dynamic light scattering and electrophoretic light scattering) in a previous work [41], where the dispersions were demonstrated to stay unchanged for at least 9 months. The average PUU particle size was found to be ca. 100 nm (in [41]).

### 3.1. Morphology

The studied PUU/silica nanocomposites were subjected to some basic morphological characterization already in the previous work [1]: light microscopy (LM) confirmed that they were transparent and smooth at the microscale; atomic force microscopy (AFM) and scanning electron microscopy (SEM) of their surface (in [1]) revealed fibril-shaped crystallites or lamellae in the neat matrix, which disappeared in the nanocomposites [1]. Additionally, silica nanoparticles and/or agglomerates in the surface- and subsurface layer were occasionally visible (SEM: backscattered electrons) in case of nanocomposites [1]. The Fourier transform infrared analysis (FTIR) completed in [1] yielded information about interfacial matrix–filler interactions (see also detailed comments in the Appendix A, Section 5 Interface interactions (FTIR)). The FTIR observations confirmed that nanosilica caused changes in hydrogen bonding, and that possibly (but not provably) the carbonate units of the elastic chains participate in matrix–filler hydrogen bonding (in addition to the strongly H-bonding urethane groups). The mentioned interactions seemed to correlate with the observed disappearance of crystallites (lamellae) in samples in which nanosilica was incorporated.

In the present work, which is dedicated to comparing macro- and micromechanical properties, the morphology analyses were focused on the dispersion pattern of silica nanospheres, which were visualized in detail by TEM and SEM. The results are summarized in Figure 1 (some more are in Appendix A in the Appendix A).

At low loadings, the silica nanospheres are not evenly distributed in the studied nanocomposites, as can be seen from the comparison in Figure 1a–d. Especially in the case of the sample PUU05 (Figure 1a), filler-rich islands can be seen embedded in a continuous filler-deficient matrix domain. At higher SiO_2_ loadings (10–20 wt%: Figure 1b,c) the filler-rich islands come closer to each other and interconnect; but even at 20% of nanosilica, its distribution still is not perfectly homogeneous, and the filler-deficient soft domains still are relatively prominent. At the highest filler loadings, 30% (not shown here, see Appendix A) and 40% (Figure 1d), the filler dispersion at the nanoscale is nearly homogeneous, and the pattern of the SiO_2_ nanoparticles practically percolates.

The global-scale morphology (observed on a fracture surface) of the nanocomposites is shown in Figure 1e–h (5 µm scale, SEM): At this magnification, the filler (bright domains) appears to be more evenly dispersed at all loadings, but fluctuations in its concentration are still visible even at this scale, especially in the case of lower filler loadings.

The observed patterns in filler distribution (i.e., the insular pattern at low loadings, and the percolating pattern at the highest filler contents) were found to play an important role in the macro- and micromechanical properties, as will be discussed further below.

### 3.2. DMTA: Macromechanical Properties

The mechanical properties of the prepared PU/silica nanocomposites at macroscale (macromechanical properties) were characterized by dynamic mechanical thermal analysis (DMTA). The results are summarized in Figure 2 (some are in Appendix A). The temperature-dependent trends observed by DMTA for the storage modulus G’, loss modulus G”, and loss factor tan(δ) (Figure 2a–c, respectively) provide interesting information about matrix–matrix and matrix–filler interactions.

The micromechanical properties, which will be discussed further below, were characterized exclusively at 25 °C (this temperature is marked by a dotted line in Figure 2). In order to compare the macro- and micromechanical properties, the values of the shear moduli (storage modulus G’ and loss modulus G”) and the damping factor (tan(δ) ) at 25 °C were extracted from the DMTA curves in Figure 2 (and in Appendix A), and are summarized in Table 2.

Detailed description and analysis of the DMTA trends with references to theory textbooks [45,48], as well as to samples studied in the author’s previous work [2,49] can be found in the Appendix A (Section “6 DMTA: Thermomechanical properties”).

To summarize the DMTA results in Figure 2, it can be noted that the curves of G’, G”, and tan(δ) display several characteristic temperature regions: the glassy region (from −80 °C to ca. −45 °C), the region of the glass transition (from ca. −45 °C to ca. −15 °C), the lower-temperature rubbery region (affected by hydrogen bonding, ranging from ca. −15 °C to ca. 100 °C), the region of abrupt melting that is found only in the samples PUU00 to PUU20 (with increasing filler loading, the mentioned melting region shifts from 108 °C to 145 °C), the high-temperature rubbery region found only in the samples PUU30 and PUU40 (from ca. 140 °C to ca. 200 °C), and, finally, the region of thermal degradation (above ca. 200 °C, which was only investigated in the samples PUU30 and PUU40).

From the trends in G’ and tan(δ), as well as from the comparisons with reference samples [2,49], it was concluded that the thermal dissociation of weak hydrogen bonds is the dominant effect in the low-temperature rubbery region (from ca. −15 °C to ca. +100 °C), while the dissociation of a stronger variety of hydrogen bonds (among hard segments of polyurethane) is responsible for the melting of the samples PUU00–PUU20. Finally, very strong hydrogen bonds (between hard segments of polyurethane and nanosilica) were suggested to be responsible for the infusibility of the highly filled samples (PUU30 and PUU40).

The temperature-dependent curves of tan(δ) in Figure 2c visualize three relaxations, all of which are well-visible in the samples PUU00–PUU20: near –34 °C, a relatively sharp peak of the glass transition is observed (very distinct in all samples), which corresponds to the relaxation of the flexible polycarbonate chains, namely of their free fraction that is not immobilized by any interactions. The same distinct relaxation was also observed in polycarbonate-based PUs in a previous work (see discussion of a polycarbonate system further below, in Section 4.2.3, second paragraph). The next relaxation peak is smaller and broader, but distinctly separate in the case of the samples PUU00–PUU20, in which it is centred at ca. 12 °C. In case of PUU20, however, the high-temperature slope of this peak already is asymmetrically up-lifted in a distinct way. At an even higher filler content, in PUU30, the second relaxation maximum is shifted to 50 °C, and it significantly overlaps with the third (and last) relaxation peak, as a shoulder on the low-temperature slope of the latter. In PUU40, the second relaxation is barely visible as a flat shoulder near 70 °C, which is nearly hidden in the slope of the broad third relaxation peak. The second relaxation was assigned to the dissociation of weak hydrogen bonds between the elastic PCD chains (soft segments) as proton acceptors, and nanosilica (Si–OH groups) as well as hard segments of the PUU matrix as proton donors. Both types of hydrogen bonds seem to be comparably strong, as just a single relaxation peak is observed, not only for the neat matrix PUU00, but also for the nanocomposites PUU05–PUU20 (and the maximum of this peak stays at the same temperature). The shifting of the maximum of the second relaxation to higher temperatures (PUU30 and PUU40) might be assigned to an increasing immobilization of the hydrogen-bonded PCD chains at high contents of nanosilica, both by the hydrogen bonding to nano-SiO_2_ (nanosilica is more abundant and more evenly distributed in these samples: see Figure 1c,d), as well as by topological exclusion due to the close spatial proximity of stiff filler nanoparticles to PCD chains (this effect is enhanced by the nearly percolating pattern in Figure 1c,d). The third relaxation, which is associated with the tan(δ) peak maximum observed between ca. 100 °C (PUU00) and 125 °C (PUU40), was assigned to the dissociation of hydrogen bonds among the hard segments of the matrix. This peak becomes broader at the highest filler contents, which will be discussed below in context of the phase mixing. This third (and last) relaxation makes possible the melting of the samples PUU00–PUU20, while the highly filled PUU30 and PUU40 stay crosslinked even after this relaxation (see G’ = f(T) in Figure 2a). The infusibility of the latter samples was assigned to very strong hydrogen bonds between hard segments and nanosilica, which do not dissociate prior to the thermal degradation of the matrix.

Considering the relatively complicated tan(δ) = f(T) curves, aspects such as the matrix–filler compatibility and the miscibility of the hard and soft segments of the polyurethane matrix are of great importance. In simple segmented polyurethanes, the micro-phase-separation of the strongly and multiply hydrogen-bonding hard segments (containing densely concentrated urethane groups)from the elastic and relatively inert soft segments plays a desired and important role, which provides strong crosslinking as well as elasticity [4,5]. A strong microphase separation was observed for polycarbonate-based polyurethanes structurally related to the presently studied ones. The mentioned related materials were prepared via catalysed room-temperature curing of a bulk mixture of precursors (see discussion of a polycarbonate PC-PU system in Section 4.2.3, second paragraph). The so-obtained microphase-separated materials displayed only one peak in their tan(δ) = f(T) curves, namely the glass transition of the free polycarbonate chains (T_g_ = –32 °C, similar like the analogous one in the presently studied PUU series). From the glass transition to the melting region, the tan(δ) = f(T) curves of the PC-PU materials discussed in Section 4.2.3 displayed an ideal plateau, and this plateau ended with a step-wise increase of tan(δ) in the melting region. Moreover, the mentioned reference samples displayed a nearly ideal rubbery plateau in their G’ = f(T) curves, and between their T_g_ and their melting point, similar to the plateau in a simple (and infusible) polymer network shown as reference in Appendix A. On the other hand, the melting region of the strongly microphase-separated samples of PC-PU materials discussed in Section 4.2.3 was very similar to the melting region of a simple semicrystalline polymer (see DMTA in Appendix A). The described course of the tan(δ) = f(T) (as well as of G’ = f(T)) curves of the microphase-separated PC-PU reference materials mentioned in Section 4.2.3 contrasts with the characteristics of the presently studied PUU series shown in Figure 2, where two additional tan(δ) relaxations at temperatures above T_g_ can be observed, and, additionally, the upward tan(δ) step in the melting region is missing. Also, the G’ = f(T) curves of the PUU series steadily decrease in the rubbery region. 

In contrast to the above-discussed strongly phase-separated segmented polyurethanes, there can be significant phase mixing between hard and soft segments if the soft ones are not inert but display a sufficient tendency for hydrogen bonding, and hence a better compatibility with the hard segments (see e.g., [27,29,35] and also the series of PC-PU materials discussed in Section 4.2.3). The preparation conditions also can play a role in the phase mixing (see e.g., waterborne polyether-based polyurethanes in [29]). Polycarbonate chains with carbonyl groups acting as proton acceptors possess a structure that can favor phase mixing in polyurethanes. In the mentioned PC-PU series of materials discussed in Section 4.2.3, one specific polycarbonate-PU matrix was prepared, which had a relatively irregular distribution of the positions of its larger hard segments. This latter PU material was found to display considerable phase mixing, and the tan(δ) = f(T) curve of this specific reference material is somewhat similar to the one observed for the presently studied PUU series: a flat and broad peak was noted in the rubbery region, in place of the plateau typical for the microphase-separated PU. Such a broad peak in the rubbery region was identified as a typical DMTA indication of PU phase mixing, also described in [29]. The phase mixing is connected with the above mentioned weak (and relatively easily dissociating) hydrogen bonds between soft segments (such as the PCD component of the PUU matrix) and hard segments. Additionally, in case of [29], and even more so in case of the phase-mixed PC-PU material discussed in Section 4.2.3, the G’ = f(T) curves of the phase-mixed polyurethanes are similar to the ones observed for the presently studied PUU series. In contrast to the mentioned literature’s examples of phase mixing in polyurethanes, the PUU series studied in this work displays one additional specific feature, namely the tan(δ) peak in the melting region, instead of a step of tan(δ). This indicates significant residual elasticity also in the molten state in all the PUU samples. It might be partly attributed to some degree of the covalent branching (such as biuret units), as well as to dynamic hydrogen bonding in the molten phase.

The compatibility of the studied PUU matrices with the employed stabilized colloid nanosilica might appear moderate in view of the morphology images of the samples with lower nano-SiO_2_ loadings in Figure 1a,b, but the DMTA results indicate that the mutual affinity of matrix and filler is considerable. The second relaxation peak in the tan(δ) = f(T) curves suggests a hydrogen bonding interaction between the soft PCD chains and the nano-SiO_2_ particles (the peak area grows with silica content); this interaction is confirmed by the growth of both G’ and G” with increasing filler content. The increasing size and width of the final relaxation peak of tan(δ) with increasing filler content additionally supports the expected high affinity of the employed nano-SiO_2_ variety to the hard segments of the PUU matrix. The latter affinity is also clearly demonstrated by the persistent crosslinking of PUU30 and PUU40 at temperatures above the third relaxation (dissociation of the crosslinking between the PUU chains via hard segments), at which range the less-filled samples (PUU00–PUU20) are molten. The suppression of crystallite lamellae by the addition of nano-SiO_2_ in the PUU series, which was observed by the authors in [1], also suggests a distinct interference of the filler with the hydrogen bonding in the matrix, and hence a good mutual compatibility.

A comparison of the trends in G’ and G” highlight that the silica nanofiller markedly raises not only the storage modulus in the rubbery region, but, interestingly, it raises the loss modulus as well. Secondly, G” is always smaller than G’, usually by somewhat more than 1 order (it is just 2 times smaller at T_g_), which means that the studied materials are always predominantly, but not ideally, elastic. The increase of both moduli were attributed to additional matrix–nanofiller hydrogen bonding, the G’ increase to the stronger variety of these H-bonds, and the G” increase to the more labile H-bonds.

The trends of all three DMTA magnitudes suggest that all the above-discussed types of hydrogen bonds occur in a wide distribution of strength. The matrix–matrix and matrix–filler hydrogen bonds are discussed in detail in Section 4.1.

### 3.3. Indentation Testing: Micromechanical Properties

Mechanical properties at the microscale (micromechanical properties) were assessed from instrumented microindentation hardness testing (MHI). Raw data in the form of typical *F*-*h* curves are shown in Figure 3. Complete results of micromechanical measurements are summarized in Table 3. The table shows final averaged values and standard deviations for all five micromechanical properties, namely: indentation modulus (*E*_IT_) proportional to macroscopic elastic modulus; indentation hardness (*H*_IT_) proportional to macroscopic yield stress; Martens hardness (*H*_M_), also known as the universal hardness; indentation creep (*C*_IT_) related to the macroscopic creep; and the elastic part of the indentation work (*η*_IT_) defined as the ratio of elastic deformation to total deformation (for details see Materials and Methods, Section 2.3.3). We note that the properties were measured with a maximum loading force of 100 mN (superscript 100, such as $E_{\mathrm{IT}}^{100}$) and 200 mN (superscript 200; $E_{\mathrm{IT}}^{200}$).

## 4. Discussion

### 4.1. Interactions in the Nanocomposites

In order to explain the trends in the macroscopic and the indentation (micromechanical) properties, the matrix–matrix and matrix–nanofiller interactions will be discussed first.

#### 4.1.1. Neat matrix: Hydrogen Bonding between Segments

Segmented polyurethanes:

Polyurethanes (PU) are characterized by extensive hydrogen bonding, which is responsible for their unique properties [4,5]. Commercially produced PUs most typically are designed to possess some elasticity [4,5]. In order to achieve this, they are synthesized as block copolymers—the segmented polyurethanes, which contain alternating hard and soft segments (blocks) [4,5] (see example in Figure 1 and detailed structure of the studied system in the Appendix A).

The soft segments, which originate in the main diol component, are typically long and flexible (and hence elastically active) polymer chains, which display either no hydrogen bonding tendency or just a weak one. The strongly hydrogen bonding hard segments (Figure 1b) are based on a sequence of short structural units that were formed from the short diisocyanate component and from the short second diol component (the chain extender). The hard segments are very rich in the hydrogen-bonding carbamate (= urethane) groups. The PUU polymer matrix studied in this work also possesses the described segmented structure: its soft segments are based on flexible polycarbonate diol (PCD), the hard ones on hexamethylene diisocyanate and 2,2-Bis(hydroxymethyl) propionic acid (DMPA; the chain extender).

The crosslinking by hydrogen bonding in segmented polyurethanes is shown in Figure 1a, for the example of the studied matrix. The hard segments are interconnected by strong hydrogen bonds (Figure 1b), which base their aggregation on nanocrystallites. The latter strongly physically cross-link (Figure 1a) the flexible linear chains (soft segments). Additionally, there can be also weak hydrogen bonding between lonely hard blocks (or their aggregates) and the soft segments, as shown in Figure 1a,c. This weak bonding (as proton acceptors) is a prominent feature of the polycarbonate soft segments. In detail, Figure 1c1 shows the stronger hard-soft interactions with C = O units of polycarbonate, while Figure 1c2 illustrates the weaker ones, with oxygen from C-O-C units, such as in the case of polyether diols in the role of the soft segment.

The effects of H-bonding in the neat matrix on its thermomechanical behavior (DMTA) is shown in Figure 2. As a result of the significantly strong soft–hard interactions between urethane and carbonate groups (Figure 1c1,c2), the studied filler-free PU matrix displays an unusually high modulus in the rubbery state (in view of the considerable length of the elastic component, M_n_ = 2870 g × mol^−1^). Additionally, in the whole rubbery region, the modulus steadily drops with increasing temperature, which can be attributed to the gradual disconnection of the soft–hard hydrogen bonds. On the other hand, the melting of the neat matrix, and also of the nanocomposites PUU05–PUU20 (see Figure 2), can be attributed to the dissociation of the hard–hard hydrogen bonds (shown in Figure 1b).

#### 4.1.2. Filler–Matrix Hydrogen Bonding

Matrix–filler interactions were found to have a very strong effect on the mechanical properties of the studied nanocomposites in the rubbery state (DMTA and microindentation results). The silica surface is covered by weakly acidic Si–OH groups which are well-suited to be proton donors for hydrogen bonding (see Figure 2). In the PU matrix of the studied nanocomposites, the soft blocks (see Figure 2a: carbonate groups as proton acceptors), as well as the hard blocks (see Figure 2b: carbamate groups as proton acceptors) are well-suited for hydrogen bonding to nano-SiO_2_. The previously observed (see author’s work [1]) suppression of larger crystallites or lamellae in the nanocomposites’ morphology, as consequence of matrix–nano-SiO_2_ interactions (observed even at 5% of SiO_2_ by AFM and DSC), also indicated an intense involvement of nanosilica in the hydrogen bonding in the studied matrix.

The hydrogen bonding between the soft blocks (PCD) and nanosilica can be assigned as the origin of most of the filler effect in the rubbery region, namely of the increase in the storage modulus (at T = const.) with increasing silica content, as observed by thermomechanical (macroscopic) analysis (DMTA: Figure 2 further above). The temperature-dependent storage modulus of the silica-reinforced nanocomposites shows the same trend of decrease with rising temperature (in the rubbery region), as does the modulus of the neat matrix itself. The rubbery moduli of the neat matrix (PUU00) as well as of PUU05–PUU20 all drop down to approximately the same final value before these mentioned samples melt. Hence, the improved rubbery moduli of the samples PUU05–PUU20 can be practically entirely (and mostly in PUU30 and PUU40) attributed to additional hydrogen bonds between soft segments and silica. These latter H-bonds seem to be of comparable strength to the H-bonding between the hard and soft segments. The H-bonding between PCD and silica can also be correlated with the up-shifting temperature of the abrupt melting of PUU00–PUU20, due to polymer chain immobilization.

The hydrogen bonding between the hard blocks (carbamate units) and silica can be correlated with the infusibility of the samples PUU30 and PUU40 (see Figure 2), and also with the higher modulus of the infusible PUU40 (in comparison to PUU30) in the second rubbery region between 120 and 200 °C. The H-bonds (crosslinks) between hard segments and nano-SiO_2_ hence must be relatively frequent in PUU30 and PUU40, and they also appear to be very strong, because they persist—the temperature range of thermal degradation is above 200 °C.

Ammonia adsorbed as a stabilizing agent on the commercial silica nanospheres:

The ammonia molecules on the surface of nano-SiO_2_ can engage in analogous hydrogen bonding, such as the unprotected Si–OH groups (Figure 2), but nevertheless they cause a significant weakening of the interfacial hydrogen bonding between the matrix and nano-SiO_2_ filler. The difference between the effect of non-stabilized vs. stabilized nanosilica was noted in the literature for polyurethane–SiO_2_ nanocomposites [30], and was observed in the case of polyacrylamide–SiO_2_ nanocomposites [50]. More detailed comments and a scheme of the surface stabilization (Appendix A) are in the Appendix A (Section: 4. Ammonia adsorbed on the silica nanospheres).

### 4.2. Correlations among Structure, Macro- and Micromechanical Properties

#### 4.2.1. Statistical Correlations between Macro- and Micromechanical Properties

Figure 4 and Figure 5 document the strong correlations between stiffness-related and energy dissipation-related properties, respectively. We note that the correlations are strong regardless of the scale, i.e., the both macro- and micromechanical properties show the same trends. This is further proof that the indentation experiments can characterize a broad range of polymer materials, including polymer nanocomposites [46].

Figure 4 summarizes the correlations between stiffness-related properties. The stiffness-related properties comprise the storage modulus *G*’ (from macroscale DMA measurements), together with the indentation modulus (*E*_IT_) and indentation hardness (*H*_IT_), both of which are attainable from microscale indentation measurements. In order to verify the reliability and reproducibility of MHI measurements, the *E*_IT_ and *H*_IT_ were measured at two different loading forces (100 mN and 200 mN), which is described in the experimental section (Section 2.3), while the measured values are shown in the results section (Section 3.3). The correlations between all properties were strong, regardless of the loading force. This confirmed that the experimental parameters implemented during the indentation experiments may influence the absolute values of the micromechanical properties, as is described elsewhere [51,52], but they usually do not change the overall trends [2,53].

Figure 5 displays the correlations between energy dissipation-related properties. The group of energy dissipation-related properties includes loss modulus (*G*”; attained from macroscale DMA measurements), together with indentation creep (*C*_IT_), and the elastic part of indentation work, *η*_IT_ (from microscale indentation measurements). Such as in the case of stiffness-related properties, the values of micromechanical properties showed the same trends regardless of the loading force (100 mN or 200 mN). With the increasing amount of filler, the loss modulus and indentation creep tend to grow, while the elasticity (expressed as the elastic part of indentation work, *η*_IT_) decreases, as documented in Table 2 and Table 3. Consequently, we observed positive linear correlations between *G*” and *C*_IT_, while the *G*”–*η*_IT_ and *C*_IT_–*η*_IT_ correlations were negative (but strong and linear).

Figure 6 quantifies the strength of all observed correlations among the mechanical properties by means of Pearson’s correlation coefficients *r*. Briefly, the Pearson’s *r* can take a range of values from 1 to −1, depending on the strength of a linear association between two variables. The value of 1 corresponds to ideal positive association (i.e., if one variable increases, the other increases as well), the value of −1 evidences ideal negative association (i.e., the increase of one variable means the decrease of the other), and the value of 0 indicates no association between the two variables. More details can be found in [54]. In Figure 6, there are several regions evidencing strong correlations among the macromechanical and micromechanical properties. We note that the figure is a heatmap symmetric with respect to the main diagonal, which contains the autocorrelations (i.e., correlation of each quantity with itself). In the upper right corner is a region of strong positive correlations between all stiffness-related properties (the range from DMA/*G*’ to MHI/*H*_M_). At the bottom and on the right, there are two symmetrically equivalent regions showing the negative correlation between *η*_IT_ and all other properties.

We can conclude that the correlations between the elasticity-related micromechanical and macromechanical properties were strong (high Pearson’s correlation coefficients *r* in Figure 6), while in case of the energy dissipation-related properties, the correlation is significant, but less marked. While Figure 4 and Figure 5 visualize the most important correlations and illustrate their linearity, Figure 6 quantifies the strength of the all correlations by means of *r*-coefficients.

The above study of the correlations between the mechanical properties and the presented analysis of the thermomechanical properties (see Figure 2: DMTA) also included the nanofiller-free matrix PUU00. At least in case of the DMTA analysis, it could be observed that the neat matrix displays analogous temperature-dependent trends in mechanical properties, such as its composites with nanosilica. This analogy in behavior can be attributed to the fact that similar types of hydrogen bonding play a key role in the thermomechanical (DMTA) properties of the neat matrix (as polyurethanes are rich in H-bonding), as well as of the nanocomposites (even more H-bonding: matrix–matrix, as well as matrix–filler). A different type of non-filled polymers, namely elastomeric epoxy networks, was previously studied by the authors in [2]: in this latter case, the macro- and micromechanical properties displayed very strong correlations, both if the elasticity-related and the energy dissipation-related properties were compared.

#### 4.2.2. Differences between Indentation Hardness and Universal Hardness

The key stiffness-related micromechanical properties—indentation modulus (*E*_IT_) and indentation hardness (*H*_IT_)—are evaluated by the *F*-*h* curves (Figure 3) in terms of the Oliver & Pharr theory (O & P theory; ref. [42]). This approach is widely accepted in the field of polymer science, although it is not 100% correct. The problem consists in that the O & P theory was developed for elastoplastic materials, while polymer systems are elastoviscoplastic [44]. This is especially true for the studied system, which at room temperature (in its rubbery plateau) displays three important properties: elasticity via the entropy spring mechanism of the molecular chains, the energy dissipation by molecular friction of the elastic chains, and the energy dissipation by the mechanically driven rearrangement of weaker hydrogen bonds. Nevertheless, for most polymer systems, the O & P theory works very well, in the sense that the measured micromechanical properties show reasonable trends and correlate with the macroscopic mechanical properties of the given systems. Moreover, it is possible to verify the applicability of the O & P theory by means of a simple test that employs universal hardness (*H*_M_), which is determined directly from the *F*-*h* curves without any theoretical assumptions [47,53]. We consider the fact that all stiffness-related properties should be proportional to each other, and then we correlate the measured values of *H*_IT_ and *H*_M_, as shown in Figure 7. If the micromechanical properties calculated in terms of the O & P theory (*H*_IT_) correlate strongly with the analogous stiffness-related micromechanical property determined directly from the *F*-*h* curves (*H*_M_), we get an indication that the values of indentation moduli (*E*_IT_) and (*H*_IT_) show correct trends, regardless of the possible energy dissipation-related effects within the studied polymer system. It is worth noting that the different absolute values of *H*_IT_ and *H*_M_ are not unusual and result not only from the energy dissipation-related effects, but also from the different definitions of *H*_IT_ and *H*_M_, as explained elsewhere [3,55].

#### 4.2.3. Macroscopic Thermo-Mechanical Properties (DMTA): Effects of Structure and Morphology

Figure 8 evaluates the trends and correlations of the macromechanical properties that were analyzed by DMTA (see Figure 2 further above), which will be now discussed in view of the matrix–matrix and matrix–nanofiller interactions (illustrated in Figure 1 and Figure 2 further above). The DMTA analyses in Figure 2 were carried out as temperature ramp tests, while microindentation tests, whose results are compared with DMTA, were carried out at T = 25 °C. Hence, in Figure 8, the DMTA magnitudes G’, G”, and tan(δ), all measured at T = 25 °C are presented and plotted as a function of filler loading in order to analyze the filler effect in DMTA in a way similar to the micromechanical tests.

A detailed analysis of the trends G’, G”, tan(δ) = f (filler loading), including references to theory textbooks [48], as well as to samples studied in previous works of the authors [1,2,49,56,57,58,59] is provided in the Appendix A (Section “7 Discussion of thermomechanical properties (DMTA): effects of structure and morphology”). Interesting was a polycarbonate-polyurethane (PC-PU) system studied in [58].

It can be seen in Figure 8 that G’ and G” display nearly identical trends if the filler amount is varied, but G” always is ca. 1 order smaller. The visibly faster growth of G’ with filler content between 30 and 40% of nanosilica was attributed to percolation of stiff filler, which supports elasticity (G’), but less so the energy dissipation-related loss modulus G”. The filler percolation also explains the significant drop in tan(δ) between 30 and 40% of silica. A notable feature is that G” is relatively high in comparison to G’ in the rubbery state of the tested materials (as noted also in the DMTA temperature ramps), which was attributed to a significant fraction of mechanically labile weak crosslinks, consisting of hydrogen bonds between soft segments on one side, and hard segments and nanosilica on the other. The curve of tan(δ) = f(T), which represents the relative tendency of energy dissipation vs. elastic energy storage (tan(δ) = G”/G’), displays a modestly changing trend between 0 and 30% of silica, which was correlated with insular (see Figure 3 further below) vs. more homogeneous filler distribution, and with labile hydrogen bonds PCD–silica (see section “7 Discussion of thermo-mechanical properties …” in the Appendix A). The error bars in the trends in Figure 8, and especially in Figure 8c (tan(δ)) are realistic: they are estimated on the basis of the statistical error analysis performed during the mechanical properties study [2], where five independent analyses yielded an averaged data point plus the corresponding error bar (such accuracy tests were done in the glassy, glass-transitioning, and rubbery state; most of the error in DMTA was found to be related to specimen fixation).

#### 4.2.4. Elastic Properties: Trends and Differences in Micro vs. Macroscopic Tests

Micromechanical indentation tests at T = 25 °C (see Figure 3 and Table 3 further above) yielded the elasticity-related magnitudes indentation modulus (*E*_IT_), indentation hardness (*H*_IT_), and Martens hardness (*H*_M_), which display nearly identical trends if compared among themselves (see Appendix A). Hence, in Figure 9 below, the trends of (*E*_IT_) are compared with those of the storage modulus G’ measured at 25 °C (macroscopic elastic property from DMTA): it can be seen that the elasticity-related magnitudes determined by the macroscopic experiment or by quasistatic microindentation follow nearly perfectly identical trends. This is in good agreement with expectation, and also with a previous work that studied nearly ideal polymer networks in different material states [2]. In the latter work, the experimentally determined ratio of *E*_IT_ to G’ comes close to the expected value for the ratio of the Young’s modulus (E) to the shear modulus (G): in the glassy polymer state, it was found to be 2.3 instead of the idealized value of 2.7, and in the rubbery state, it was equal 2.4 instead of the idealized 3.0.

In the case of the presently studied PUU nanocomposites, the ratios of *E*_IT_ to G’ are relatively far from the expected values, in contrast to the simple polymer networks from [2]: while for ideal rubbery materials the ratio *E* = 3 G’ is expected, we can observe in Figure 9 (as well as in Table 2 and Table 3) for the PUU series that the experimental *E*_IT_ is approximately equal to the experimental G’ (PUU40: E = 1.1 G’ or 1.3 G’, depending on the applied indentation force of 100 or 200 mN, respectively), or *E*_IT_ is even smaller than G’ (pure matrix: E = 0.54 G’ or 0.58 G’, with 100 or 200 mN, respectively). This finding can be explained by the already mentioned reversible nature of much of the crosslinking at T = 25 °C (H-bonds to PCD) in the studied samples. In contrast to the covalent networks studied in [2], the presently studied nanocomposites can display cold flow upon large deformations (dissociation/reconnection of crosslinks shown in Figure 1c1,c2 and in Figure 2a), but at the same time they do not creep at small deformations and lightly applied stresses (similar to the weakly crosslinked materials studied in [59]). This explanation is further supported by the fact that the mismatch between the expected and found *E*/G’ ratio becomes less dramatic if the nanocomposites are more strongly physically crosslinked, namely at higher loadings of the nanosilica reinforcement.

#### 4.2.5. Energy-Dissipation-Related Properties: Trends and Differences in Micro vs. Macroscopic Tests

Micromechanical indentation tests (see Figure 3 and Table 3 further above) yielded the following energy dissipation-related properties (the indentation creep (*C*_IT_), and the elastic part of indentation work (*η*_IT_), respectively). These magnitudes are nearly ideally complementary, so that *C*_IT_ ~ (100—*η*_IT_), as can be seen in Appendix A, and *C*_IT_ is the magnitude that directly quantifies the energy dissipation via permanent (creep) deformation. Macroscopic mechanical properties, which are directly related to energy dissipation, are the loss modulus G”, as is also true for the loss factor tan(δ). In Figure 10, the trends (dependent on nanofiller loading) of the micro- and macromechanical energy dissipation-related properties are compared. It can be observed that the trend of the indentation creep (Figure 10a) is significantly different from the trends of both the magnitudes determined by DMTA (Figure 10b,c). This is in contrast to the results obtained recently by [2] for simple chemically crosslinked polymer networks, which were tested in different material states, ranging from glassy, via glass transition, to rubbery. In that previous work, the micro- and macromechanically determined energy dissipation-related properties correlated very well (especially G” and *C*_IT_). As was discussed above, the trends of both the energy dissipation-related DMTA magnitudes G” and tan(δ) strongly differ from each other for the PUU series.

It should be noted that the differences in the trends in Figure 10 are statistically valid (also between *C*_IT_ and tan(δ) ), as highlighted by the error bars: the average values and the error bars of *C*_IT_ were obtained from 30 independent indentations. In the case of the DMTA magnitudes, the error bars are estimated on the basis of the statistical error analysis performed in [2], where five independent analyses yielded an averaged data point plus the corresponding error bar (such accuracy tests were done in the glassy, glass-transitioning, and rubbery state; most of the error in DMTA was found to be related to specimen fixation).

For a tentative explanation of the strongly different trends in the energy dissipation-related properties in Figure 10, the internal interactions (hydrogen bonding), nanofiller dispersion, as well as the cold flow behavior at different deformation (studied in [59]) can be considered. Additionally, molecular friction, and the disconnection and recombination of weak hydrogen bonds (which is connected with cold flow or creep) have to be considered as separate energy-dissipating effects. A detailed discussion is provided in the section “9 Energy-dissipation-related indentation properties …” in the Appendix A.

In summary, three regions of filler content can be discerned in Figure 10: 0 to 5 (10 wt%), 5 to 20 wt%, and 20 to 40 wt% of SiO_2_. Important effects contributing to the differences in trends are small vs. large local deformations (the latter associated with cold flow), the suppression of crystallite lamellae by nano-SiO_2_ (as observed in [1]), labile hydrogen bonds, as well as morphology effects, such as even vs. uneven (insular: see Figure 3) distribution of the nanofiller. In the first region, the growth of *C*_IT_ was assigned to the suppression of crystallites and to large local deformations, the growth of G” to an increasing number of labile hydrogen bonds, and the decrease of tan(δ) to the presumably less easily yielding insular morphology combined with small local deformations endured in DMTA. In the second region, the drop in *C*_IT_ was assigned to the increasingly strong effect of the insular morphology (meaning better elasticity at large local deformations), while G” growth and the increase in tan(δ) were assigned to an increasing number of labile hydrogen bonds. In the third region, the growth of *C*_IT_ was assigned to the disappearance of the insular structure (homogeneous filler pattern) and thus to lower deformations needed for cold flow, while local deformations during indentation were relatively high. The growth of G” again was assigned to the increasing number of labile hydrogen bonds, and the decrease of tan(δ) to the filler percolation effect combined with small local deformations endured in DMTA.

## 5. Conclusions

The aim of this work was to compare the macro- and micromechanical (indentation) behavior of novel rubbery polymer nanocomposites, which displayed rich and diverse hydrogen bonding in their internal structure.

The studied materials are elasto-visco-plastic at room temperature, in contrast to metals, for example, which are just elastoplastic.

The nanocomposites were based on a poly(urethane urea) PUU matrix and on highly regular, nearly monodisperse silica nanospheres (20–30 nm) as the filler, with filler loadings ranging from 0 to 40 wt% (0 to ca. 21 v/v%).

In the microscale, the samples were characterized at room temperature via quasistatic microindentation hardness testing (MHI), which yielded elasticity-controlled magnitudes, such as indentation modulus, indentation and Martens hardness, and energy dissipation-related magnitudes, such as indentation creep and indentation work. The macroscopic mechanical properties were characterized by the dynamic mechanical thermal analysis (DMTA) in a relatively wide temperature range, thus yielding the temperature dependent storage and loss moduli (and also their tan(δ) ratio).

The elasticity-related properties, i.e., the storage shear modulus (macroscopic, DMTA), as well as the indentation modulus and hardness (both MHI), were evaluated as functions of the nanofiller loading. Their trends were nearly identical, and the statistical correlation was strong (all Pearson’s correlation coefficients *r* > 0.9 and corresponding *p*-values < 0.001).

The energy dissipation-related properties displayed fairly different trends and only moderate statistical correlation, not only in case of comparing indentation vs. DMTA, but also if loss modulus vs. loss factor (both DMTA) were comparable. Only the theoretically expected complementary character of the micromechanical magnitudes of indentation creep and indentation work were confirmed.

The complex trends in energy dissipation-related properties were highly affected by the existence of hydrogen bonding of broadly varied strength, as revealed by the distribution patterns of the fine nanofiller, as well as by eventual locally-endured larger deformations. Cold flow at moderately large local deformations (mechanical disconnection of the weakest H-bonds) was suggested to be at the origin of the anomalies in the trends of the energy dissipation-related properties, as well as of the strong deviation of the ratio of indentation modulus to the macroscopic shear modulus. Generally, molecular friction and the disconnection and subsequent recombination of weak hydrogen bonds (cold flow under energy dissipation) manifested themselves as separate effects, so that the information obtained from the different energy dissipation-related magnitudes was not mutually equivalent, but additive.

## Data Availability

Data are contained within the article.

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
