# Peer review of "Morphology, Micromechanical, and Macromechanical Properties of Novel Waterborne Poly(urethane-urea)/Silica Nanocomposites"

_materials, 2023, doi:10.3390/ma16051767_

Round 1

Reviewer 1 Report

Very interesting report that studied the relation between macro and micro mechanical measurements (and properties) in PUU nanosized Silica nanocomposites. This report includes a thorough introduction, interesting DMTA data, clear discussion of interactions at the molecular level (though only qualitative), and discussion of correlations between different measurements.  Although microindentation measurements are beyond my expertise, the authors explain the procedure, assumptions, and results well. Overall the report very good. It is a bit long, but detailed enough to keep the attention of interested readers. My recommendation is that it should be published with minor edits. 

1) please clearly define the following terms: 

- "waterborne" in the introduction 

- "O&P" in the introduction 

- "MHI" in the introduction

- Each parameter in line 398 by name

2) I do not understand the second bullet point of page 8, beginning on line 338. do the authors mean -30 to 100 C? Otherwise, please clarify what they are claiming here. Perhaps include a citation to "typical rubbers"

3) Given the language of peaks in figure 2c shifting in the form of a shoulder on the melting peak, please describe in more detail what the authors believe is the origin of this observation. I do not believe it has to do with melting and don't know if that is (or what is) being implied. Please indicate where this is discussed in the interactions section. 

4) regarding figure 6, can we assume there is a linear dependence or is looking solely at correlations better? (this is a question of X increases linearly with y vs X increases as Y increases). What would happen without NPs in unfilled PUU systems? Or what about random polymeric materials? 

5) The authors claim that because G' nad G" trends are similar as a function of NP loading, they are driven by the same physio-chemical effect (line 686). While possible, I do not think this is the only explanation and caution the authors on asserting such conclusions. In general, this section was a bit unconvincing to me. 

6) why is Figure 9 split into two graphs as opposed ot plotted on the same axes to show the "nearly perfect identical trends"? If that is true, it would be better to show that more clearly on the same plot. 

7) in general, the authors can improve their arguments adn the readers understanding (throughout section 4) by including citations to relevant literature which they must have used to draw such conclusions. Otherwise, the data is not thorough enough to definitively draw such mechanistic conclusions. 

8) I found the discussion around Figure 10 lacking. I also question the statistical validity of measurement trends in Figure 10a (are the error bars std dev of a single measurement or many different fabricated samples? perhaps I missed that detail...). The authors go on to analyze this trend which I think should be removed or edited. 

9) I recommend the authors reduce the text of the manuscript, especially and starting with the conclusion. Certain parts of the discussion could also be reduced.  

Author Response

Response to comments of Reviewer #1

Reviewer #1:

Very interesting report that studied the relation between macro and micro mechanical measurements (and properties) in PUU nanosized Silica nanocomposites. This report includes a thorough introduction, interesting DMTA data, clear discussion of interactions at the molecular level (though only qualitative), and discussion of correlations between different measurements.  Although microindentation measurements are beyond my expertise, the authors explain the procedure, assumptions, and results well. Overall the report very good. It is a bit long, but detailed enough to keep the attention of interested readers. My recommendation is that it should be published with minor edits. 

Answer: The authors are grateful for the positive assessment, for the Reviewer’s attention paid to the submitted work, and for the subsequent valuable suggestions, which helped to significantly improve the Manuscript and make it more attractive and reader-friendly.

The authors attach a copy of the revised Manuscript with highlighted changes as a separate file.

Important changes done in response to the suggestions and questions of Reviewer #1 were highlighted in yellow; response to points suggested by both reviewers: magenta; responses to the other reviewer - Reviewer #2: turquoise.

Reviewer #1:

1) please clearly define the following terms: 

- "waterborne" in the introduction  (p.2)

- "O&P" in the introduction  (p.2)

- "MHI" in the introduction  (p.3)

- Each parameter in line 398 by name (p.9)

Answer: The authors defined and explained the mentioned terms at the suggested places (now in the revised Manuscript: lines: 59, 94/95, 138, and 455–460, respectively).

Reviewer #1:

2) I do not understand the second bullet point of page 8, beginning on line 338. do the authors mean -30 to 100 C? Otherwise, please clarify what they are claiming here. Perhaps include a citation to "typical rubbers"

Answer: The mentioned problematic text (“Between –30 °C and –100 °C, an …”) on line 338 on page 8, was corrected to “Rubbery region: In the range from –30 °C to +100 °C, an …” in (now line 351 on page 8). The authors are grateful for the attentive reading and apologize for this typographic error.

Also, the whole section “3.2. DMTA: Macromechanical properties” was carefully checked for reader friendliness, streamlined and improved (strongly improved or added texts are highlighted).

A citation describing moduli of “typical rubbers” was added (Mezger T G: The Rheology Handbook For users of rotational and oscillatory rheometers, 4th ed.; Vincentz Network: Hanover, Germany, 2014, ISBN13: 978-3-86630-650-9). Also, as a response to point (7) of Reviewer #1, all statements of similar type like above were provided with corresponding citations.

Reviewer #1:

3) Given the language of peaks in figure 2c shifting in the form of a shoulder on the melting peak, please describe in more detail what the authors believe is the origin of this observation. I do not believe it has to do with melting and don't know if that is (or what is) being implied. Please indicate where this is discussed in the interactions section.

Answer: The whole section “3.2. DMTA: Macromechanical properties” was carefully checked for reader friendliness (also as part of response to point (2)), streamlined and improved (the strongly improved or added texts are highlighted), including the discussion of the “shifting peaks” in Figure 2c (initially small separate peaks for neat matrix and low-filled nanocomposites, eventually shoulders on neighbouring large peak in highly filled materials – this is now much better explained).

Reviewer #1:

4-first part)    regarding figure 6, can we assume there is a linear dependence or is looking solely at correlations better? (this is a question of X increases linearly with y vs X increases as Y increases).

Answer: Figure 6 is a standard statistical plot (the correlation matrix table in the form of heatmap). The purpose of this plot is to summarize and quantify the strength of correlations between multiple pairs of quantities, especially if the number of quantities is so high that standard XY-plots would take too much space and would not fit into one page. Nevertheless, the reviewer is right Pearson’s correlation coefficient r do not discriminate linear and monotonic relations. However, the previous Figures 4 and 5 show the most important correlations and prove that they are mostly linear, while Figure 6 adds the information about the strength of all observed correlations. We slightly modified (highlighted text: lines 679–681 and 685–687 in the revised work) the paragraph describing Figure 6 so that this context is clearer.

Reviewer #1:

4-second part)    What would happen without NPs in unfilled PUU systems? Or what about random polymeric materials?

Answer: This is an interesting topic. In the mentioned section (discussion of indentation properties and their correlations with macromechanical ones), the authors added some new comments at lines 694-705 and 708/9 in the revised work, as answer to this point.

Generally, one specimen with no NPs was characterized in the presented work: it is the neat matrix PUU00. It is only one data point in the indentation studies (w(filler) = 0%), as these could not be conducted at different temperatures. But the data point PUU00 well fits into the trendlines of indentation magnitudes, so that its behaviour can be considered to be very similar to the nanocomposites. Also in the temperature-dependent DMTA tests, PUU00 and its nanocomposites show similar-shaped curves. This similarity of trends for matrix and nanocomposites is not automatic. But in the presented case, the matrix contains very rich and diverse (several types) hydrogen bonding. The equally hydrogen-bonding nanofiller only additionally participates in the existing diverse H-bridging, without changing the overall character of the material behaviour.

Simpler “random polymeric materials” also were included into the commented Results description, as well as into the Discussion. For this purpose, the additional SI-Figure 3 was added to the Supplementary Information File. It includes DMTA profiles (sets of curves G’, G”, tan_delta = f(T) ) for four reference materials, which were synthesized or studied as specimens in the authors’ previous work (relevant citations also were added). These materials include a simple semicrystalline linear PCL polymer (with Tg and melting point), a simple rubbery epoxy network (which was studied in a previous work “citation [2]” by the authors; the work was dedicated to micro vs. macro-mechanics), as well as materials with reversible physical and chemical crosslinks. The presently studied materials (PUUxx) display the most complicated behaviour if compared to the “references”, because of the complexity of the PUU matrix. Comparisons with the mentioned reference materials were also drawn to support discussion and conclusions elsewhere.

Reviewer #1:

5) The authors claim that because G' and G" trends are similar as a function of NP loading, they are driven by the same physio-chemical effect (line 686). While possible, I do not think this is the only explanation and caution the authors on asserting such conclusions. In general, this section was a bit unconvincing to me.

Answer: The commented statement indeed was unfortunate. While the authors wanted to highlight the importance of hydrogen bridges for both elasticity and viscosity, a statement in the sense that “trend analogy of G’ and G” automatically implies a common physico-chemical origin of both trends” was not appropriate. In fact, the post-glass-transition course of G’ and G” displays analogous trends in most types of simple polymeric materials, as can be seen in the above-mentioned newly added SI-Fig. 3 in the Supplementary Information File.

The discussion in the section “4.2.3. Structure and morphology effect on dynamic-mechanical properties (DMTA)” at some places was not reader friendly, also. The authors thoroughly checked this section and strongly improved some of its paragraphs (highlighted).

The mentioned SI-Fig. 3 was added to help the discussion by means of comparison with four different polymeric materials (“references”), simple as well as complex ones, which were previously tested by the authors.

As a special feature, it is noted that in the whole rubbery region, G” is anomalously high if compared to G’. This is a strong difference in comparison to simple (not composite) polymeric materials, but it is a similar feature shared with materials from SI-Fig. 3 which display thermally reversible crosslinks. A fifth “reference” material, from citation [58] in the revised text (not shown in SI-Fig. 3 but only discussed), possesses weak physical crosslinks and easily undergoes shear liquefaction via mechanical disconnection of these crosslinks. This latter material in many experimental situations showed similarly high ratio of G” to G’ in the rubbery region like the studied PUU materials.

The drop (instead of step-wise increase) of tan(delta) upon melting of the meltable products PUU00 to PUU20 is another unusual feature of the studied materials in comparison to the simple ones among the set in SI-Fig. 3.

The section “4.2.3. Structure and morphology effect on … DMTA)”was strongy revised, better structured and improved by mentioning the above aspects (including important literature citations). The postulated key role of the H-bridges in the high values of both G’ and G” is debated in this context.

Reviewer #1:

6) why is Figure 9 split into two graphs as opposed to plotted on the same axes to show the "nearly perfect identical trends"? If that is true, it would be better to show that more clearly on the same plot.

Answer: The graphs were joined into one graph as suggested by the Reviewer.

Reviewer #1:

7) in general, the authors can improve their arguments and the readers understanding (throughout section 4) by including citations to relevant literature which they must have used to draw such conclusions. Otherwise, the data is not thorough enough to definitively draw such mechanistic conclusions.

Answer: The reviewer is right. The authors apologize for this shortcoming. The authors carefully revised all the text of Results and Discussion and added the missing citations of handbooks, basic scientific works (hydrodynamic effect of filler), as well as of works documenting own specific experience (partly new citations, partly cross-citations to the previously existing literature list). All conclusions and discussions were verified according to point (7). In total, 7 new citations were added. Also, the mentioned SI-Figure 3 with DMTA data of four reference materials was added.

Reviewer #1:

8) I found the discussion around Figure 10 lacking. I also question the statistical validity of measurement trends in Figure 10a (are the error bars std dev of a single measurement or many different fabricated samples? perhaps I missed that detail...). The authors go on to analyze this trend which I think should be removed or edited. 

Answer: The differences in the trends in Figure 10 are realistic, as well as the error bars:

-The average values and error bars of CIT were obtained from 30 independent indentations (Section Materials and Methods).

-In case of the DMTA magnitudes, there was indeed only a single measurement per graph in the present study, but the error bars are estimated using the experience from statistical error analysis performed during the study [2], where five independent DMTA analyses yielded an averaged data point plus the corresponding error bar – such accuracy tests were done in the glassy, glass-transitioning, and rubbery state. (Most of the error in DMTA was found to be related to specimen fixation: this was seen by comparison of result scattering, if a single sample was measured five times without removal from clamps, vs. the measuring the same piece of specimen five times but removing and fixing again each time, vs. five different specimens from the same material, fixed and removed individually).      The validity of the error bars is now stressed in the revised Manuscript.

Because of the validity of the error bars, the observed trends had to be interpreted by the authors. (With markedly larger error bars, the trends of G” and tan_delta, for example, indeed would be identical within the error margin).

The original discussion of the Fig. 10 was not very reader friendly. It was much improved, streamlined and better arranged in the revised Manuscript. Also, some conclusions drawn here were supported by experience from the newly cited work [58], which was dedicated to elastomers with weak physical crosslinks, which easily underwent shear liquefaction via mechanical disconnection of crosslinks, at deformations higher than a few percent.

Reviewer #1:

9) I recommend the authors reduce the text of the manuscript, especially and starting with the conclusion. Certain parts of the discussion could also be reduced.

Answer: The Conclusion section was thoroughly revised and shortened by half, as suggested by Reviewer #1. Next, the Abstract was much revised and shortened too. The sections Introduction, Results, as well as Discussion were thoroughly checked for reader-friendliness, streamlined and shortened. However, added explanatory texts requested by both Reviewers, as well as added literature references, caused that the revised Manuscript has a very similar length like the original.

Reviewer 2 Report

The authors developed a new polyurethane polyurea/silica nanocomposite based on polyurethane, which was prepared by mixing with nano-silica aqueous suspension latex. Customizable products ranging from stiffer elastic polymers to semi-glass products are obtained. Interestingly, the authors compared the macroscopic and microscopic mechanical behavior of polymeric nanocomposites, the relationship between macroscopic and microscopic mechanical properties is also analyzed, but there are still some minor problems in the paper, which can be accepted after modification.

1. The author gives the transmission image of polyurethane silica composite, which is based on the characterization of the composite emulsion. However, there is little information on the characterization of the polyurethane silica composite emulsion, which makes people doubt the stability of the composite emulsion. It is suggested to add the basic characterization of the composite emulsion such as storage stability or mechanical stability.

2. It is recommended to supplement FTIR to confirm the successful preparation of polyurethane silicon composites. It can be used as supporting information.

3. In Figure 2a, with the increase of silicon content, the energy storage modulus gradually increases at the same temperature. Please explain the reason for this phenomenon.

4. In Figure 5, when the viscosity of the material is high, its loss modulus and indentation creep tend to increase, while the elasticity decreases. Please explain the reason for this phenomenon.

5. It is suggested that the conclusion of the paper should be appropriately condensed.

6. The author should polish the grammar of the paper thoroughly.

Author Response

Response to comments of Reviewer #2

Reviewer #2:

The authors developed a new polyurethane polyurea/silica nanocomposite based on polyurethane, which was prepared by mixing with nano-silica aqueous suspension latex. Customizable products ranging from stiffer elastic polymers to semi-glass products are obtained. Interestingly, the authors compared the macroscopic and microscopic mechanical behavior of polymeric nanocomposites, the relationship between macroscopic and microscopic mechanical properties is also analyzed, but there are still some minor problems in the paper, which can be accepted after modification. 

Answer: The authors are grateful for the positive assessment, for the Reviewer’s attention paid to the submitted work, and for the subsequent valuable suggestions, which helped to significantly improve the Manuscript and make it more attractive and reader-friendly.

The authors attach a copy of the revised Manuscript with highlighted changes as a separate file.

Important changes done in response to the suggestions and questions of Reviewer #2 were highlighted in turquoise; response to points suggested by both reviewers: magenta; responses to the other reviewer - Reviewer #1: yellow.

Reviewer #2:

  1. The author gives the transmission image of polyurethane silica composite, which is based on the characterization of the composite emulsion. However, there is little information on the characterization of the polyurethane silica composite emulsion, which makes people doubt the stability of the composite emulsion. It is suggested to add the basic characterization of the composite emulsion such as storage stability or mechanical stability.

Answer: The TEM images, which depict the silica nanofiller, in fact were taken in the solvent-free state, using specimens of dry nanocomposites, not frozen emulsions. The stability of the “PUU” emulsion, which was a precursor of the presently studied final dry product, nevertheless is an interesting topic.

For more clarity, the authors added some explaining text into the revised Introduction (presently on lines 61–65, 67, and 77), as well as an introducing first paragraph in the section Results (presently on lines 273–283), which briefly summarizes how the studied materials were obtained prior to the mechanical tests (namely according to a recent publication by the authors – citation [1] in the revised Manuscript). The added text also reports the stability of the employed polyurethane dispersion: at least 9 months. The stability was studied by the authors in the citation [36] (as numbered in the revised Manuscript), via Dynamic Light Scattering, and via Zeta Potential determination by Electrophoretic Light Scattering.

Reviewer #2:

  1. It is recommended to supplement FTIR to confirm the successful preparation of polyurethane silicon composites. It can be used as supporting information.

Answer: As the here-studied nanocomposites were for the first time prepared in a recent work by the authors [1], the filler­-matrix interactions were already studied by FTIR in this latter paper. Because hydrogen bridging (which is conveniently characterized by FTIR) plays a very important role also in the mechanical properties studied in the present work, the authors cite the conclusions from the FTIR analyses in the revised Manuscript (lines 292–298): changes in hydrogen bridging of the urethane groups, and, possibly, but not provably also in bridging to the carbonate units. Some more detailed comments about FTIR are included into the section 2 of the Supplementary Information File. The newly added citation [57] on line 797 (p. 21, section “4.2.3. Structure and morphology effect on dynamic-mechanical properties (DMTA)”) additionally mentions the proof by FTIR of the important hydrogen bridging to the ‘soft’ polycarbonate segments in a partly similar polycarbonate-based nanocomposite.

Reviewer #2:

  1. In Figure (2a), with the increase of silicon content, the energy storage modulus gradually increases at the same temperature. Please explain the reason for this phenomenon.

Answer: The discussion of the nature of reinforcement by nano-silica in context of Fig. 2a (Results section) was expanded and modified (p.8/9, lines 360–366), and a new citation was added with this discussion ([47]).

The assignment of the reinforcing effect (which increased with silica content and hence with nanofiller surface) to hydrogen bridging polyurethane–silica was done with the help of a comparison with the expected hydrodynamic effect of a rigid nanofiller according to [47].

Reviewer #2:

  1. In Figure 5, when the viscosity of the material is high, its loss modulus and indentation creep tend to increase, while the elasticity decreases. Please explain the reason for this phenomenon.

Answer: We suppose that the reviewer wanted to explain the sentence in the paragraph just below Figure 5, which reads: “If a material exhibits higher viscosity, the loss modulus and indentation creep tend to grow, while the elasticity (ηIT) decreases [42].” If this is the case, the reviewer is right that the term “viscosity” is a bit confusing here and requires further explanation. As the detailed discussion of structure-viscosity-properties relationships is given in the following subsection 4.2.4, we decided to simplify the ambiguous sentence as follows: “With the increasing amount of filler, the loss modulus and indentation creep tend to grow, while the elasticity (expressed as the elastic part of indentation work, ηIT) decreases, as documented in Tables 2 and 3.”   (lines 662–664 in the revised Manuscript)

Reviewer #2:

  1. It is suggested that the conclusion of the paper should be appropriately condensed.

Answer: The Conclusion section was thoroughly revised and shortened by half, as suggested by Reviewer #2 (and also by Reviewer #1). Next, the Abstract was much revised and shortened too. The sections Introduction, Results, as well as Discussion were thoroughly checked for reader-friendliness, streamlined and shortened. However, added explanatory texts requested by both Reviewers, as well as added literature references, caused that the revised Manuscript has a very similar length like the original.

Reviewer #2:

  1. The author should polish the grammar of the paper thoroughly.

Answer: The authors politely disagree with the mention that grammar itself was problematic. However, the authors admit, that the stylistics of some parts of the manuscript was not reader friendly. The authors thoroughly checked the whole Manuscript for reader friendliness, streamlined and they shortened and simplified the discussion texts. During the revision, the authors paid a great attention to replacing complex sentences and expressions by reader friendly ones.
